# The Embryotoxic Effects of in Ovo Administered Sunset Yellow FCF in Chick Embryos

**DOI:** 10.3390/vetsci8020031

**Published:** 2021-02-18

**Authors:** Fatma Colakoglu, Muhammet Lutfi Selcuk

**Affiliations:** 1Department of Nutrition and Dietetics, Faculty of Health Sciences, Karamanoglu Mehmetbey University, 70100 Karaman, Turkey; 2Department of Physiotherapy and Rehabilitation, Faculty of Health Sciences, Karamanoglu Mehmetbey University, 70100 Karaman, Turkey; mlselcuk@kmu.edu.tr

**Keywords:** chick, embryotoxic effect, liver, kidney, sunset yellow

## Abstract

Sunset yellow (SY) at prescribed concentrations has been approved by regulatory authorities in several countries as an additive dye in the food, beverage, cosmetic, and pharmaceutical industries. However, there are some reports that it may cause several health problems. The aim of this study is to evaluate embryotoxic effects of SY on liver and kidney in chick embryos. Babcock white Leghorn eggs were randomly divided into four groups. Non-treated eggs served as control group. The eggs in groups SY_200_, SY_1000_, and SY_2000_ were treated with a single injection of 200, 1000, and 2000 ng SY into the air sac just before incubation. The developmental stages of embryos were determined on the 10th, 13th, 16th, and 21st days of incubation. Samples of the liver and kidney were taken and routine histological procedures were performed. The highest relative embryo weight was seen in all SY treated groups on the 16th day of incubation. Necrosis of some hepatocytes and cytoplasmic degenerations were observed in all SY groups in the liver. There were degenerated or destructed renal cortex structures and necrosis in the kidney. The cell’s nuclear areas and diameters of renal cortex structures were different in all SY groups compared to the control group (*p* < 0.05). It was concluded that in ovo administered SY has many unfavorable effects on liver and kidney in chick embryos. The results obtained in this study suggest that it may be advisable to re-assess safety levels of SY in many industries.

## 1. Introduction

People prefer ready-to-eat foods as a result of changing living conditions [1]. The use of food dyes that enhance the attractiveness of food and beverages occupies an important place in the food industry [2]. Although some azo dyes have been identified as carcinogens in research studies, it is seen that they are widely used in the food industry [3]. Sunset yellow (SY), an azo dye, is commonly used as a colorant in many ready-eat-foods, cosmetics, and pharmaceutical industries [4]. This dye is the sulfonated version of Sudan I. In the obtained product at the end of production, the presence of Sudan I is a cause of its carcinogenic effect. It is also reported that the consumption of SY causes hypersensitivity and allergic immune responses in sensitive individuals to certain substances such as aspirin [5,6]. Furthermore, it is said that this dye leads to the formation of aggressive behavior in children [7].

The average daily consumption (ADI) of this dye has undergone many changes over the years. In 1982, ADI was accepted as 0–2.5 mg/kg (body weight, BW) by the FAO/WHO Expert Committee on Food Additives (JECF). After, this rate was changed to 0–4 mg/kg [8]. Although it is also said that this dye has no adverse effects when its limit is not exceeded, recent studies show that this rate is exceeded by the manufacturers [2].

Jonnalagadda et al. [9] examined the type, size, and colorants of the sold in ready-to-eat food on the street of the 545 samples. They found that the permissible limits were exceeded in 73% of the samples, and they were reported that tartrazine and SY were the most commonly used food dyes.

When other studies are examined, it is seen that the information about the embryotoxic effects of SY is quite limited. These reasons are the relatively low limit value of this substance, metabolization in the liver, kidneys, and intestines, insufficient information about their passage rates through the placenta in mammals. Since fertilized chick eggs do not have placental barrier, they are considered as the most favorable test material for determining the effects of physical and chemical factors on organs during the embryonic period. The purpose of this study is to evaluate the embryotoxic effects of SY on development of the liver and kidney.

## 2. Materials and Methods

### 2.1. Eggs, Embryos, and Treatment Groups

Babcock white Leghorn eggs (*n* = 160) were bought from the commercial company (Alternatif poultry, Konya, Turkey) for this study. The eggs were disinfected with fumigation process in the incubator. For fumigation, a mixture of 40 mL formalin (40%) and 20 g potassium permanganate is used for per cubic meter. The disinfection was carried out for 30 min [10]. After, randomly selected eggs were placed in a hatching machine as control, SY_100_, SY_200_, and SY_2000_ groups. (temperature: 37.4–37.6, humidity: 55–65%). Eggs were opened on the 10th, 13th, 16th, and 21st days of incubation considering organogenesis. Some researchers report that the number of eggs used for each dose is very important in terms of the reliability of the results obtained from embryotoxicity studies [11,12]. Considering both the reliability of the results and the fact that some hatching eggs may be infertile, we allocated 10 eggs to each group. On the appointed days, eggs were randomly opened from each of these groups until six live embryos were obtained.

### 2.2. SY Dosing and Incubation

SY application doses were calculated according to limit (4 mg/kg/BW), determined by JECFA. (LD_50_ dose) as ADI [8]. All eggs were disinfected from the blunt ends with 96% ethyl alcohol and treated in the air sacs with a single injection of 200, 1000, or 2000 ng SY (CAS 2783-94-0; 90% purity, Sigma-Aldrich, Darmstadt, Germany) dissolved in 20 μL of bi-distilled water. It is multiplied by 10^–2^ to adapt the dilution concentration of the toxic dose determined by chicken embryotoxicity screening test (CHEST). The obtained value was accepted as the toxic dose per kilogram of maternal body weight in pregnant mammals. SY doses were calculated with consideration of this multiplier in this study. The holes were made with an egg-piercer tool (Eierpicker, Humbert and Brandts, Berlin, Germany) and immediately sealed with liquid paraffin after the injection. Eppendorf Research Plus sterile-tipped micropipette (Merck, Darmstadt, Germany) was used for all injections.

### 2.3. Macroscopic Evaluation of Embryos

On the appointed days of incubation, the embryonic developmental stages of the alive and dead embryos were defined according to the Hamburger and Hamilton Scale [13]. Alive embryos and vitellus sac weights were weighed with a precision scale (Ohaus Pioneer, PA224C, LabGear, CA, USA). The relative embryo weights were estimated by dividing the embryo weight to pre-hatching egg weight. 

### 2.4. Collection and Processing of Tissue Samples

On the appointed days of incubation, eggs were opened from each group until 6 alive embryos were obtained. Liver and kidney tissues samples were taken from 96 chick embryos. These tissues were fixed in 10% neutral-buffered formalin solution, dehydrated in graded alcohol series, cleared in xylene, and embedded in paraffin blocks for histological examinations. Serial sections were taken at regular intervals (5 µm thickness) from these tissues blocks. These sections were stained with Crossmon’s trichrome staining [14], hematoxyline and eosin [15], and modified May-Grünwald-Giemsa staining [16]. These histological preparations were examined with a light microscope (Leica DM-2500 attached to a DFC-320 digital camera, Leica microsystems, Wetzlar, Germany) for structural changes and abnormalities.

### 2.5. Evaluation of the Stained Tissue Samples

In the assessment of the nuclear areas in different regions of the liver sections, 25 hepatocytes having nuclei were evaluated [17]. In addition, the nuclear areas of the structures (glomerular capillary endothelial cells, proximal and distal tubules) in cortex region of the kidney were measured for 25 cells with nuclei [18]. The 6 glomeruli, proximal and distal tubules diameters, and 6 renal corpuscular and glomeruli areas were also measured in renal sections [19]. The measurements were analyzed with an ImageJ Analysis Programme [20].

### 2.6. Statistical Analysis

Statistical analyses were performed using the SPSS software version 21 (IBM SPSS, New York, USA). The variables were investigated using visual (histograms, probability plots) and analytical methods (Kolmogorov–Smirnov/Shapiro–Wilk’s test) to determine whether or not they are normally distributed. Data are expressed as means ± standard error (SE). The one-way analysis of variance (ANOVA) was used to compare data obtained from the study. Levene test was used to assess the homogeneity of the variances. *p* < 0.05 was accepted as statistically significant. When an overall significance was observed, pairwise post-hoc tests were performed using Tukey’s test.

## 3. Results

### 3.1. Developmental Changes

On the appointed days of incubation, the embryonic developmental stages of the alive and dead embryos in the all groups were determined according to the Hamburger and Hamilton [13]. The embryonic deaths of the SY_200_ group had occurred in the 17th–19th days of incubation whereas the most of the embryos had death in 8th–9th days of incubation in the SY_2000_ group. In the SY_1000_ group, some embryos died on the 13rd and 17th days of incubation. However, when examined eggs that were opened on certain days of incubation, embryonic developmental retardation was also observed in some embryos according to the incubation stage in the SY_1000_ group according to the Hamburger and Hamilton Scale [13].

When the vitellus sac weights were compared, there was no statistical difference among all groups (*p* > 0.05). The daily egg weight loss was in intervals 0.53%–0.69%. In evaluation of the relative embryo weights, the highest values were seen in the SY_1000_ and SY_2000_ groups when compared to the control group on the 16th day of incubation (*p* < 0.001) (Table 1).

### 3.2. Histopathological Changes

The histopathologic assessment of the liver sections was given in Figure 1. Microscopically, control group showed normal histological structure. In different embryonic periods of incubation, histopathological changes were different in the SY treated groups when compared with the control group. These changes were more severe in the SY_2000_ group than the other SY treated groups. The observed changes in SY treated groups were disruptions in hepatic cords, dilatation and congestion in hepatic sinusoids, cytoplasmic degenerations, Kupffer cell accumulation, and inflammatory cell infiltration.

In the evaluation of the nuclear areas of hepatocytes in different regions of the liver sections, on the 10th, 13th, and 16th days of incubation, it was observed that hepatocytes in the SY_2000_ group had the smallest nuclear area when compared to the other groups (*p* < 0.05). On the 21st day, while there was no statistically significant difference in nuclear areas of all SY treated groups, the smallest value was also found in the SY_2000_ group. SY_200_ and SY_1000_ groups did not differ statistically in all embryonic periods (*p* > 0.05) (Table 2).

The microscopic evaluation of the kidney sections was given in Figure 2. The normal kidney histology was observed in the control group. In the different embryonic periods of incubation, SY treated groups showed various histological differences. In histological examination, degenerative structural disorders of glomeruli and renal tubules were determined in varying degrees according to SY doses and embryonic periods. Degenerated and/or destructed glomeruli and renal tubules, congestion of glomerular tufts and intertubular regions, hemorrhage, atrophied or distorted glomeruli, infiltration of inflammatory cells surrounding degenerated or distorted glomeruli and renal tubules, vacuolar degeneration, dilated renal tubules, renal fibrosis, and necrosis were noted in SY treated embryos.

In the measurements of the nuclear areas of glomerular capillary endothelial cells in cortex region of the kidney, the smallest nuclear areas were in the SY_2000_ group on the 10th day of incubation (*p* < 0.05). The SY_1000_ group was statistically similar to SY_2000_ group. On the 13th and 16th days of incubation, there was no difference among all the groups (*p* > 0.05). On the 21st day of incubation, the smallest nuclear areas were in the SY_2000_ group.

In the evaluation of nuclear areas of the proximal tubule cells in the renal cortex, whereas there was no statistical difference among all the groups, the smallest nuclear areas were in the SY_2000_ group on the 10th, 13th, and 16th days (*p* > 0.05). On the 21st day, the SY_2000_ group had the smallest nuclear areas when compared to the other groups. Furthermore, the SY_1000_ group was similar to the SY_2000_ group (*p* < 0.05).

In the evaluation of nuclear areas of the distal tubule cells in the renal cortex, SY_2000_ group had the smallest nuclear areas in the all embryonic periods. Furthermore, SY_1000_ group was similar to the SY_2000_ group (*p* < 0.05).

In the measurements of glomerulus and renal corpuscular areas, the smallest glomerulus and renal corpuscular areas were in the SY_2000_ group when compared to the other groups throughout all embryonic periods. On the 13th and 21st days of the incubation, all the SY treated groups were statistically different than the control group, and their values were smaller than that of the control group (*p* < 0.05) (Table 3).

In the evaluation of proximal and distal tubules diameters in the renal cortex, all the SY treated groups had the largest proximal and distal diameters when compared to the control group whereas there was no statistical difference among the groups on the 10th and 13th days of the incubation (*p* > 0.05). On the 16th and 21st days of incubation, the largest renal tubules diameters were found in the SY_2000_ group (*p* < 0.05). Furthermore, the SY_1000_ group showed similarity the other SY treated groups (*p* < 0.05).

## 4. Discussion

SY is a food dye that is approved for use in the food, beverage, cosmetic, and pharmaceutical industries in many countries [21,22,23]. Several studies report that it may cause various health problems [22,23], although they do not clarify whether these problems are due to toxicity of SY per se or to the product being used at non-approved concentrations. The results of this study may help to clarify the issue.

In the present study, according to the Hamburger and Hamilton [13] scale, embryonic deaths were observed in the incubation periods in the SY_2000_, SY_1000_, and SY_200_ groups, respectively. The earliest embryonic deaths were observed in the SY_2000_ group. However, some embryos died on the 13rd and 17th days of incubation in the SY_1000_ group. When examined eggs that were opened on certain days of incubation, embryonic developmental retardation was observed in some embryos according to the incubation stage in the SY_1000_ group. Finally, these data demonstrate an indication that SY is embryotoxic effect. The results obtained from this study were in agreement with results of Joshi and Pancharatma [24].

In embryotoxicity studies, incubation conditions should be optimal for the reliability of the study. The lost weight by evaporation of water from the egg pores is important in determining the relative (%) humidity of the incubator. This rate should be at 0.55–0.70% intervals throughout incubation [25]. In the present study, the daily egg weight loss was at intervals 0.53–0.69% (Table 1). The obtained rates were within the recommended values. Additionally, this situation indicates that our incubation conditions are optimal conditions. Further, embryo weights of the control groups were found 3.11, 8.91, 22.74, and 41.18 g in incubation on days 10th, 13th, 16th, and 21st, respectively. According to Table 1, these values were agreement with the data of Romanoff [26]. Observed differences might be due to egg size and/or chick breed.

In evaluation of the relative embryo weights, the highest values were found in the SY_1000_ and SY_2000_ treated groups when compared to the control group on the 16th day of incubation (*p* < 0.001). Although there was no statistical difference, the vitellus sacs of the SY_2000_ group embryos were found larger than the other groups, especially on the 16th and 21st days of incubation. This finding showed that the embryos could not sufficiently benefit from the vitellus. Consequently, these data might be considered as an indication of developmental retardation in embryos.

In order to maintain homeostasis in the organism, the liver and kidneys are important organs in many physiological and functional mechanisms such as maintaining fluid-electrolyte balance, regulating blood pressure, detoxification, blood production, hormonal regulation and storage. Especially in intoxication, both of these organs play a significant role when exposed to the toxins, chemicals, and plant extracts. Compounds of toxins absorbed by intestinal cells may be hepatotoxic and renal toxic effect [27]. Obtained results indicated that treating embryos with SY caused significant histopathological changes in the liver. These changes are possible cytotoxic activities of SY [28]. In comparison with the control group, the observed histological changes were evident in the liver sections of SY treated chick embryos according to the dose and embryonic period. Even at the lowest dose of SY, serious histopathological effects were observed in both liver and kidney tissues.

In the microscopic evaluation of liver and kidney tissues, the most serious structural and morphological changes were determined in the SY_2000_ group when compared to the other SY treated groups for all embryonic periods of incubation. It was observed that the cytotoxic effect of dye was more advanced according to the doses and progressed embryonic periods. Disorganized hepatic cords, dilated hepatic sinusoids and central veins, congestion, and inflammatory cells infiltration were observed in all embryonic periods of all the SY treated groups. Kupffer cell activation began to be seen in the 16-day-old embryos of the SY_200_ group, while this activation had already occurred in the some 10-day-old embryos of the SY_2000_ group. When compared to the other SY groups, the most severe degenerations were in the 21-day-old chick embryos of the SY_2000_ group (Figure 1). Histological changes in kidney tissue were inflammation, hemorrhage foci, necrosis, vacuolization, degenerated or/and destructed renal cortex structures, deterioration of renal corpuscular integrity, dilatation of Bowman’s space, atrophied glomeruli, thickened Bowman’s capsule, dilated renal tubules, and congestion (Figure 2). These histopathological findings were similar to the data of some studies with SY [27,28,29,30].

The presence of reactive oxygen species showing oxidative stress caused in the hepatic and renal tissues of the toxic metabolites of SY or/and the inhibition of the antioxidant defense mechanism of the toxic metabolites interacting with the proteins and enzymes of both tissues may cause these histopathological changes [28,29]. Frances et al. [31] reported that oxidative stress might occur due to an increase in the level of the reactive oxygen species, a deterioration in the normal antioxidant system or both. Some studies suggest reactive oxygen species disrupt membrane integrity and permeability. Water and sodium ions as a result of the increased permeability are caused intensively transferred into the cells. Finally, serious cytotoxic effects were caused in the liver and kidney tissues [32,33]. Cemek et al. [34] reported that azo dyes are metabolized by azoreductase in the liver.

In all the SY treated groups, seen vacuolar and hydropic degenerations due to increased intracellular fluid indicate liver and kidney damage [35]. In both of the tissues of SY treated embryos, the accumulation of the inflammatory cells shows the initiation of an immune response against inflammation caused by dye metabolites [30,36,37]. The marked increase of inflammatory cells is highly apparent in the 21-day-old chick embryos of the SY_2000_ group in both tissues.

In this study, it was seen that the numbers of hepatic sinusoidal Kupffer cells increased progressively depending on the dose and exposure duration of SY. The numbers of Kupffer cells increase in phagocytosis of accumulated SY metabolites. Kupffer cell hyperplasia is also associated with the degree of hepatic tissue injury caused by SY intoxication [38]. In this study, observed necrosis of hepatocytes and renal tubules occurs after the structural deterioration of cell organelles, shrinking and dissolution of nuclei [39]. The p53 gene known as a tumor suppressor plays an important role in many functions such as cell cycle, cell differentiation, inflammation, and immunity. Especially, it is a potent transcription factor [40]. This gene shows a protective effect against DNA damage in cells when cells are exposed to genotoxic stimulants. If damaged DNA is to be repaired, p53 activates other genes related to telomere maintenance, DNA repair, centromere structure, and telomere deficiency. However, if DNA cannot be repaired, the p53 gene causes to drift into apoptosis of the cell by preventing cell division. [41]. Based on this information, it is understood that SY metabolites that react with DNA may also cause nucleus mutations [35].

In the present study, the vascular alterations observed in the various tissues are an indication of SY induced endothelial injury. Coagulation factors may have been deteriorated by SY leading to a lack of coagulation. Congestion occurs as a result of slowing blood flow, inadequate venous return, and dilatation of arterioles [42]. Furthermore, the reason of the observed vascular changes is focal dilatations [35]. Hyaline degeneration may occur in small muscular arteries and arterioles. This degeneration may occur in fibrous tissue in the scars and some conditions due to the deposition of glycoproteins between collagen bundles. It may also occur in the walls of blood vessels [43]. Hydropic degeneration of hepatocytes and vacuolation of structures in the renal cortex indicate the presence of intracellular edema, as a result of toxicity or immune aggressions [44]. Intact glomerulus membrane is important for normal glomerular infiltration rate. Proteinuria is an outcome of degenerated and disrupted glomerular structure [30,45]. Pathologically, fibrosis is the accumulation of connective tissue. Renal fibrosis occurs as a result of inflammation and oxidative stress in the progressive kidney diseases [46,47]. Obtained data were similar to the results of some studies with azo dyes [27,30,48].

In the present study, when evaluating the nuclear areas of hepatocytes and structures in the renal cortex, all the SY treated groups were statistically different than the control group, and their values were smaller than that of control group (*p* < 0.05). Especially, the smallest nuclear areas were in the SY_2000_ group throughout all embryonic periods. The SY_1000_ group was statistically similar to the SY_2000_ group. Generally, SY_200_ and control groups did not differ statistically. In the measurements of glomerulus and renal corpuscular areas, the smallest glomerulus and renal corpuscular areas were in the SY_2000_ group. In addition, the other SY treated groups were smaller than that of the control group. Shrinkage of nuclear area causes decreasing protein synthesis. Shrinkage of nuclear areas forms as a result of decreased protein synthesis due to reducing the number of ribosome [21]. If DNA of the nucleus exposed to genotoxic stimulants cannot be repaired, the p53 gene prevents cell division. Addditionally, the damaged cell goes to apoptosis [41]. Furthermore, the reason for shrinkage in the glomerulus and renal corpuscular areas is cell necrosis according to obtained measurements from this study.

When evaluating proximal and distal tubules diameters in the renal cortex, the largest renal tubules diameters were in the SY_2000_ group. Furthermore, SY_1000_ group showed similarity the other SY treated groups. Renal tubule dilatation may occur anywhere along the nephron or collecting duct system. This situation may result from toxic injury in the tubule epithelium interfering with absorption and secretion [49]. In addition, we thought that SY metabolites might cause to an epithelial injury with the present study.

## 5. Conclusions

Based on the results obtained in this study, it was concluded that SY, which is commonly used as colorant in the food, beverage, cosmetic, and pharmaceutical industries, has deleterious effects on the developing chick embryo when administered in ovo. The effects on liver and kidney were proportional to the dose given. Therefore, we suggest that the current safety levels of SY be re-evaluated, particularly in relation to changes in the embryonic period. In addition, we consider that our procedure may be the basis of a suitable method to include embryonic effects when determining the safety of a food additive. 

## Figures and Tables

**Figure 1 vetsci-08-00031-f001:**
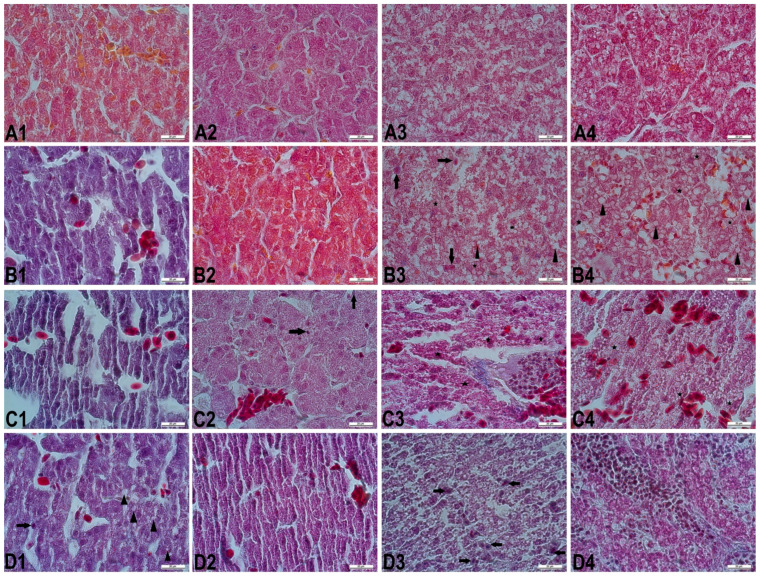
The liver sections in different embryonic periods. (**A1**–**A4**): Normal histological structure in the control group. (**B1**–**B4**): The liver sections from chick embryos treated with SY_200_. (**B1**): Slightly disorganized hepatic cords, dilatation, and congestion in hepatic sinusoids (day 10). (**B2**): Slightly irregular hepatic cords (day 13). (**B3**): A few Kupffer cells (arrows), disorganized hepatic cords, marked hidropic degeneration (stars) in hepatocytes, and mildly vacuolar degeneration (arrowheads) with small droplets (day 16). (**B4**): Moderately dilatation and congestion in hepatic sinusoids, and marked vacuolar degeneration (arrowheads) with small and large droplets, and hidropic degeneration (stars) in hepatocytes (day 21). (**C1**–**C4**): The liver sections from chick embryos treated with SY_1000_. (**C1**): Severe disorganized hepatic cords, dilatation in hepatic sinusoids and central vein, and hyperemia (day 10). (**C2**): Dilated central vein with erythrocytes, and some hepatocytes having pkynotic nuclei (arrows) (day 13). (**C3**): Diffuse disorganized hepatic cords, Kupffer cells, marked hyaline degeneration (stars) in hepatocytes, and massive infiltration of mononuclear cells in portal vein (day 16). (**C4**): Severe congestion in hepatic sinusoids and blood vessels, hepatocytes having irregularly shaped nuclei, and diffuse marked hidropic degeneration (stars) (day 21). (**D1**–**D4**): The liver sections from chick embryos treated with SY_2000_. (**D1**): Kupffer cell (arrow), slightly vacuolar degeneration (arrowheads) with small and large droplets, sinusoidal extensions, slightly impaired hepatic cords (day 10). (**D2**): Dilated central vein with erythrocytes, and diffuse hyaline degeneration in hepatocytes (day 13). (**D3**): Kupffer cells (arrows), necrosis of most hepatocytes, hepatocytes having irregularly shaped nuclei (day 16). (**D4**): Severe marked hyaline degeneration in hepatocytes, massive infiltration of inflammatory cells in portal region, and necrotic foci (day 21). Crossmon’s trichrome staining. Bar = 20 µm.

**Figure 2 vetsci-08-00031-f002:**
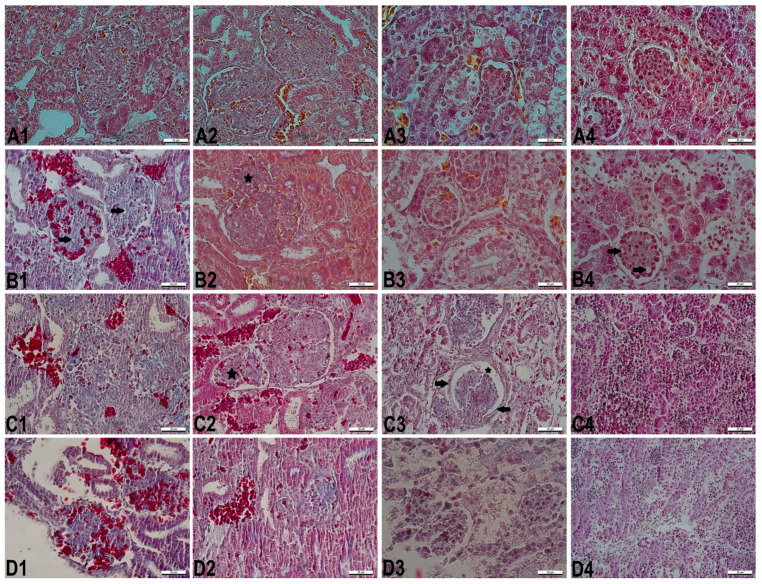
The kidney sections in different embryonic periods. (**A1**–**A4**): Normal histological structure in the control group. (**B1**–**B4**): The kidney sections from chick embryos treated with SY_200_. (**B1**): Spills in the renal tubules epithelium, vacuoles in glomeruli (arrowheads), congestion of glomerular tufts, intertubular hemorrhage foci, and periglomerular leukocyte infiltration (day 10). (**B2**): Congestion of glomerulus and intertubular blood vessels, and diminished glomerulus (star) (day 13). (**B3**): Severe spills in the renal tubules epithelium, swelling in some tubule epithelial cells, and slightly hyperemia in the intertubuler region (day 16). (**B4**): Marked vacuoles in glomeruli tufts (arrows), degenerated and destructed renal tubules, marked nuclear disorders, necrosis of most tubular epithelium (day 21). (**C1**–**C4**): The kidney sections from chick embryos treated with SY_1000_. (**C1**): Highly degenerated and distorted glomeruli/renal tubules, intertubular hemorrhage foci, and inflammatory cells infiltration in renal cortex (day 10). (**C2**): Slightly congestion of glomeruli, severe intertubular hemorrhage, atrophied glomerulus (star), and dilated tubules (day 13). (**C3**): Thickened parietal layer of Bowman’s capsule (arrows), dilatation of Bowman’s space (star), diminished glomerulus, degenerated tubules, and marked fibrosis in the glomerular tufts (day 16). (**C4**): Highly massive infiltration of inflammatory cells in renal cortex region (day 21). (**D1**–**D4**): The kidney sections from chick embryos treated with SY_2000_. (**D1**): Highly massive hemorrhage in glomerular tufts and intertubular region, and partially degenerated renal tubules (day 10). (**D2**): Atrophied and destructed glomerulus, degenerated tubules, and massive hemorrhage (day 13). (**D3**): Degeneration and destruction of most tubules, deterioration of renal corpuscular integrity, and tubular necrotic foci (day 16). (**D4**): Highly massive infiltration of inflammatory cells in renal cortex region, degeneration and destruction of most renal tubules (day 21). Crossmon’s trichrome staining. Bar = 20 µm.

**Table 1 vetsci-08-00031-t001:** Some morphometric values of the embryos used in the study (Mean ± SE).

Day	Group	First Egg Weight (g)	Pre-hatching Egg Weight (g)	Embryo Weight (g)	Relative Embryo Weight (%)	Daily Egg Weight Loss (%)	Vitellus Sac Weight (g)
10th	C	60.71 ± 2.58	56.51 ± 2.61	3.11 ± 0.09	5.50	0.69	1.91 ± 0.16
	SY_200_	57.68 ± 1.54	54.61 ± 1.28	3.10 ± 0.17	5.68	0.53	1.82 ± 0.17
	SY_1000_	58.54 ± 1.96	54.78 ± 4.62	3.26 ± 0.14	5.95	0.64	1.77 ± 0.12
	SY_2000_	57.82 ± 2.40	54.46 ± 2.36	3.21 ± 0.16	5.89	0.58	2.27 ± 0.38
13th	C	58.03 ± 2.25	53.06 ± 2.12	8.91 ± 0.19	16.79	0.65	1.23 ± 0.04
	SY_200_	58.44 ± 0.87	53.94 ± 2.19	8.79 ± 0.56	16.30	0.59	1.52 ± 0.18
	SY_1000_	59.36 ± 1.19	54.38 ± 1.19	8.98 ± 0.37	16.51	0.64	1.48 ± 0.16
	SY_2000_	58.03 ± 1.37	53.61 ± 1.58	8.96 ± 0.49	16.71	0.58	1.58 ± 0.12
16th	C	58.58 ± 1.13	52.78 ± 1.57	22.74 ± 1.11	43.08 ^a^	0.61	2.13 ± 0.18
	SY_200_	56.48 ± 1.82	50.59 ± 1.98	24.68 ± 1.38	48.78 ^ab^	0.65	2.13 ± 0.36
	SY_1000_	55.44 ± 0.84	49.50 ± 0.98	26.12 ± 0.53	52.77 ^b^	0.66	2.60 ± 0.31
	SY_2000_	56.81 ± 2.04	50.88 ± 1.82	25.39 ± 0.82	49.90 ^b^	0.65	2.63 ± 0.34
21st	C	56.07 ± 1.53	49.01 ± 0.91	41.18 ± 0.92	84.02	0.60	2.18 ± 0.32
	SY_200_	60.67 ± 1.42	53.05 ± 1.65	44.89 ± 1.89	84.62	0.60	2.89 ± 0.66
	SY_1000_	57.67 ± 1.48	50.67 ± 1.77	37.85 ± 2.96	74.70	0.58	3.39 ± 0.65
	SY_2000_	58.00 ± 1.49	50.59 ± 1.49	40.76 ± 1.42	80.57	0.61	4.35 ± 0.74

^a–b^ Values within a column with no common superscripts are significantly different (*p* < 0.001).

**Table 2 vetsci-08-00031-t002:** Nuclear areas of structures in liver and kidney tissues (Mean ± SE).

Day	Group	Nuclear Areas of Hepatocytes (µm^2^)	Nuclear Areas of Glomeruli (µm^2^)	Nuclear Areas of Proximal Tubules (µm^2^)	Nuclear Areas of Distal Tubules (µm^2^)
10th	C	20.40 ± 0.61 ^c^	23.97 ± 0.74 ^c^	21.61 ± 0.42	20.05 ± 0.36 ^b^
	SY_200_	16.25 ± 0.52 ^b^	24.31 ± 0.85 ^c^	20.21 ± 0.99	18.33 ± 0.49 ^ab^
	SY_1000_	14.10 ± 0.52 ^b^	19.39 ± 1.81 ^ab^	20.49 ± 0.49	18.19 ± 0.85 ^ab^
	SY_2000_	9.70 ± 0.66 ^a^	17.68 ± 1.06 ^a^	19.18 ± 0.22	16.58 ± 0.61 ^a^
13th	C	22.30 ± 0.49 ^b^	21.07 ± 1.19	22.37 ± 0.48	21.74 ± 1.18 ^b^
	SY_200_	17.20 ± 1.01 ^b^	17.67 ± 3.14	20.99 ± 0.22	17.71 ± 0.85 ^a^
	SY_1000_	15.45 ± 0.52 ^b^	16.32 ± 1.64	21.39 ± 2.32	16.82 ± 0.58 ^a^
	SY_2000_	8.40 ± 0.62 ^a^	16.40 ± 1.63	18.47 ± 1.11	16.79 ± 0.18 ^a^
16th	C	19.55 ± 0.58 ^c^	9.92 ± 0.48	14.87 ± 0.31	13.39 ± 0.12
	SY_200_	15.25 ± 0.55 ^b^	10.06 ± 0.39	15.64 ± 0.22	13.74 ± 0.66
	SY_1000_	16.00 ± 0.67 ^b^	8.52 ± 0.70	15.50 ± 0.46	13.35 ± 0.69
	SY_2000_	13.10 ± 0.44 ^a^	9.56 ± 1.02	15.12 ± 1.18	13.09 ± 1.25
21st	C	21.45 ± 0.77 ^b^	7.43 ± 0.72 ^b^	12.18 ± 0.56 ^b^	10.55 ± 1.21 ^b^
	SY_200_	16.05 ± 0.77 ^a^	8.69 ± 0.64 ^b^	12.31 ± 1.27 ^b^	11.54 ± 0.91 ^b^
	SY_1000_	14.10 ± 0.84 ^a^	8.33 ± 1.28 ^b^	10.69±1.93 ^ab^	7.89 ± 1.55 ^ab^
	SY_2000_	14.05 ± 0.55 ^a^	4.74 ± 0.29 ^a^	6.82±0.15 ^a^	6.13 ± 0.35 ^a^

^a–c^ Values within a column with no common superscripts are significantly (*p* < 0.05) different.

**Table 3 vetsci-08-00031-t003:** The measurements of areas and diameters in the kidney structures (Mean ± SE).

Day	Group	Areas of Glomeruli (µm^2^)	Areas of Renal Corpuscules (µm^2^)	Diameters of Proximal Tubules (µm)	Diameters of Distal Tubules (µm)
10th	C	10911.18 ± 358.62	15508.41 ± 595.25	52.85 ± 1.73	36.39 ± 3.93
	SY_200_	9928.72 ± 1184.10	13476.77 ± 1407.93	55.83 ± 1.83	38.62 ± 1.65
	SY_1000_	8201.13 ± 1294.55	12401.27 ± 1354.53	54.11 ± 1.18	44.65 ± 3.26
	SY_2000_	7965.46 ± 883.15	11306.88 ± 1741.95	56.12 ± 1.41	47.14 ± 2.58
13th	C	15411.71 ± 534.25 ^b^	21981.23 ± 890.81 ^b^	58.39 ± 2.61	37.43 ± 3.42
	SY_200_	10670.66 ± 1171.59 ^a^	14507.59 ± 1501.94 ^a^	57.87 ± 5.78	42.53 ± 2.71
	SY_1000_	10188.73 ±1190.86 ^a^	13945.97 ± 1468.99 ^a^	52.03 ± 1.97	42.91 ± 1.66
	SY_2000_	9644.33 ± 1121.13 ^a^	13529.14 ± 1260.89 ^a^	63.19 ± 0.91	47.76 ± 2.31
16th	C	2131.26 ± 1218.45	3738.86 ± 2267.49	19.93 ± 1.57 ^a^	18.53 ± 0.84 ^a^
	SY_200_	1927.21 ± 1089.76	3390.74 ± 2054.15	27.98 ± 1.65 ^b^	21.74 ± 4.18 ^ab^
	SY_1000_	840.98 ± 84.23	1384.33 ± 129.65	34.39 ± 1.69 ^bc^	28.43 ± 3.22 ^ab^
	SY_2000_	814.75 ± 74.96	1326.86 ± 106.88	36.99 ± 2.83 ^c^	30.67 ± 2.27 ^b^
21st	C	1154.01 ± 63.77 ^b^	1960.04 ± 119.33 ^b^	27.98 ± 1.25 ^b^	21.22 ± 1.05 ^a^
	SY_200_	888.64 ± 92.24 ^a^	1448.81 ± 90.06 ^a^	26.59 ± 1.64 ^b^	24.65 ± 0.73 ^a^
	SY_1000_	897.31 ± 126.11 ^a^	1357.56 ± 94.44 ^a^	25.79 ± 1.53 ^ab^	26.26 ± 0.53 ^ab^
	SY_2000_	863.39 ± 68.45 ^a^	1294.52 ± 165.85 ^a^	28.73 ± 2.05 ^a^	29.48 ± 1.97 ^b^

^a–c^ Values within a column with no common superscripts are significantly (*p* < 0.05) different.

## Data Availability

Not applicable.

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
