# Peer review of "The Embryotoxic Effects of in Ovo Administered Sunset Yellow FCF in Chick Embryos"

_vetsci, 2021, doi:10.3390/vetsci8020031_

Round 1
Reviewer 1 Report
The scope of the present study was to verify the effects of SY, a colorant substance in chick embryos. The design of the study is well done and the results are well described and they are of the interest in food industry. In the same manner, the discussion is quite linked. However, in relation to the conclusions of the work, as follows " It may be concluded the use of products containing SY 361 throughout pregnancy carries great risks for the health of both mother and foetus", is incorrect if extrapolated to mammals. In fact, as as well stated by the authors, chick eggs do not have placental barrier. On the other hand, it is fairly known that placenta offers the essential interface for exchange of oxygen, nutrients, and metabolic wastes between mother and fetus. Moreover, placenta is capable of both Phase I and Phase II biotransformation of many different types of drugs, xenobiotics, and endogenous compounds through the use of diverse enzymes that are capable of oxidation, reduction, and conjugation reactions. Thus, it is not possible to extrapolate the effects of the toxicity of a given compound on a fetus of birds, assuming these same toxic effects on a fetus of mammals. In fact, the studies of toxicity of compounds in conducted in JECFA must be in mammal species (e.g rats), being accepted other animal species (e.g chicken) as an accessory or complementary evaluation.
Thus, my suggestion is that the conclusions must be completely reformulated.
Author Response
Dear Reviewer,
Thank you very much for your valuable comments. We also agree with the information you express. The sentences we used in the conclusion of our study may have been expressed incorrectly. In this study, we aimed to see the effects that the permissible dose of sunset yellow determined by the JECFA could shape in a living being without placental barrier. We think that this study can be a model for studies that can be done in many mammal species, including humans. We aim to contribute to the studies of researchers who are considering working on this subject by presenting our data.
As a result of your valuable comments, we rearranged the conclusion part of our study.
Best regards
Reviewer 2 Report
The current research investigated the toxicity of SY to chicken embryos. The manuscript is written vey well with clear descriptions of methods and the interpretation of the results.
Although authors provided some explanation of the benefits of using chicken embryo as an animal model to test the toxicity of a chemical, more citations are required to support the conclusion that “the reliability of this daily consumption limit should be reassessed according to the obtained results from present study” stated in Line 356-358. Similarly, as authors in the conclusion stated that “It may be concluded the use of products containing SY throughout pregnancy carries great risks for the health of both mother and fetus.” in line 361-352, more citations should be provided to support the relationship between the chicken embryo studies and the actual human responses in the Introduction and/or discussion.
Line 360-361, delete the space
Author Response
Dear Reviewer,
Thank you very much for your valuable comments. We aimed to write the sentences we used in the conclusion of our study as a recommendation. I think we may have misrepresented this situation. Although there are not many studies on the effects of sunset yellow on embryo development, we would like to share all the information we can find with this study. In this study, we aimed to see the effects that the permissible dose of Sunset yellow determined by JECFA can shape in a living (chick embriyo) being without placental barrier. We think that this study can be a model for studies that can be done in many mammal species, including humans. We aim to contribute to the studies of researchers who are considering working on this subject by presenting our data. As a result of your valuable comments, we rearranged the conclusion part of our study.
Best regards
Reviewer 3 Report
See attached file.

Author Response
Dear Reviewer,
Thank you very much for your valuable comments. All the issues you have identified in your report have been carefully examined and evaluated. The necessary corrections have been made by monitoring (as track changes) the changes in the word file. We think that this study can be a model and a reference for other researchers who plan to work on this subject. We are grateful to you for your contribution to our study.
Please see the attachment for the point-by-point response.
Best regards

Round 2
Reviewer 1 Report
In this 2nd version of the manuscript we had detected many alterations on the text. It was really improved; however, as this study performed in chick embryo and not in mammals I missed a discussion of the results showed here, with those of SY in developing mammals (e.g. Tanaka T. Reproductive and Neurobehavioral Effects of Sunset Yellow FCF Administered To Mice in the Diet. Toxicology and Industrial Health. 1996;12(1):69-79), and many others, including those of developmental studies for evaluation of SY in JECFA. The authors advocate that considering the data observed in the present study, JECFA should to reassess the ADI obtained previously to SY as follows "....however, we think that the reliability of this daily consumption limit might be reassessed according to the obtained results from present study....". On the other hand, in my point of view they did not present yet data enough from other studies in mammals that could support theirs assumption.
Author Response
Thank you for your valuable comments on improving our article.
The aim of this study is to evaluate the embryotoxic effects of SY on the liver and kidney in chick embryos. The article "Tanata" you recommend is on Reproductive and Neurobehavioral Effects in mice. There are many articles investigating the effect of SY on other systems. However, since we aimed to evaluate the effects on the liver and kidney in terms of embryotoxic, literature in this direction was included in our study.
Toxicity studies of compounds carried out at JECFA, in studies conducted in mammalian animal species (rodents) as a complementary assessment when SY doses are passed through the placental barrier, or when it is considered that the immune systems of mammals are more resistant than humans, the use of chick eggs without a placental barrier in these studies is more meaningful. . We think that this study can be a model in terms of its effects and placental barrier that can develop in many mammalian species, including humans.
Our opinion to re-evaluate the recommended daily dose of JECFA is a recommendation given in the conclusion section. Considering the purpose of your study, we do not think it is necessary to rearrange the discussion.